# Estimating Expected Calibration Error for Positive-Unlabeled Learning

**Ryuichi Kiryo**                                                    *kiryo@ms.k.u-tokyo.ac.jp*
*The University of Tokyo, Japan*

**Futoshi Futami**                                          *futami.futoshi.es@osaka-u.ac.jp*
*The University of Osaka, Japan*
*The University of Tokyo, Japan*
*RIKEN AIP, Japan*

**Masashi Sugiyama**                                                    *sugi@k.u-tokyo.ac.jp*
*RIKEN AIP, Japan*
*The University of Tokyo, Japan*

**Reviewed on OpenReview:** *https://openreview.net/forum?id=SvoBtLIrPZ*

## Abstract

The reliability of probabilistic classifiers hinges on their calibration—the property that their confidence accurately reflects the true class probabilities. The expected calibration error (ECE) is a standard metric for quantifying the calibration of classifiers. However, its estimation presumes access to ground-truth labels. In positive-unlabeled (PU) learning, only positive and unlabeled data are available, which makes the standard ECE estimator inapplicable. Although PU learning has been extensively studied for risk estimation and classifier training, calibration in this setting has received little attention. In this paper, we present PU-ECE, the first ECE estimator for PU data. We provide non-asymptotic bias bounds and prove convergence rates that match those of the fully supervised ECE with an optimal bin size. Furthermore, we develop an information-theoretic generalization error analysis of PU-ECE by formalizing the conditional mutual information (CMI) for a PU setting. Experiments on synthetic and real-world benchmark datasets validate our theoretical analysis and demonstrate that our PU-based ECE estimator achieves performance comparable to that of the fully-labeled ECE estimator.

## 1 Introduction

Deep learning models achieve state-of-the-art accuracy in various classification tasks (Devlin et al., 2019; Dosovitskiy et al., 2021; Radford et al., 2021). However, their deployment in safety-critical applications requires more than just high prediction accuracy. For instance, in medical diagnosis (Jiang et al., 2012), autonomous systems (Muthali et al., 2023), and financial decision-making (Bahnsen et al., 2014), the confidence of the predictions provided by machine learning models must accurately reflect the true probability of the predicted events. This property, known as *calibration* (Dawid, 1982; Guo et al., 2017), is essential for reliable decision-making. The calibration error, or *true calibration error* (TCE) (Naeini et al., 2015; Guo et al., 2017; Roelofs et al., 2022), is a widely used metric for evaluating the calibration of classifiers. The TCE is typically estimated by partitioning the confidence score space into bins, followed by the calculation of the average difference between the confidence score and the empirical probability of the true label given the confidence score in each bin (Zadrozny & Elkan, 2001; Naeini et al., 2015). The resulting estimator is called the *expected calibration error* (ECE)[1] (Roelofs et al., 2022). Theoretical analysis of the ECE has also been

---

[1]As some researchers have pointed out, both TCE and ECE are sometimes called "ECE". For clarity, we follow the convention of Futami & Fujisawa (2024).

conducted, establishing bias bounds and the consistency of the ECE to the TCE as the number of samples increases, with an appropriate binning strategy (Gupta & Ramdas, 2021; Futami & Fujisawa, 2024). The results show an $\mathcal{O}(n^{-1/3})$ convergence rate of the ECE, where $n$ is the number of test samples. The naive computation of the ECE requires a fully labeled dataset, where each sample has both confidence scores and true labels.

However, in practice, fully labeled data may not be available or may be costly to obtain (Jaskie & Spanias, 2019; Li et al., 2022; Bepler et al., 2019). For instance, when identifying protein particles in microscopy images, annotating negative data is costly due to diverse backgrounds (Bepler et al., 2019). In such cases, positive-unlabeled (PU) learning, which aims to train a classifier from only positive and unlabeled data, is a promising approach (Liu et al., 2002; Elkan & Noto, 2008; Natarajan et al., 2013; du Plessis et al., 2014).

Despite extensive research on PU learning methods, the majority of these methods have focused on maximizing classification accuracy (Elkan & Noto, 2008; du Plessis et al., 2015; Kiryo et al., 2017; Chen et al., 2020; Xinrui et al., 2023). On the other hand, calibration in the PU setting has been largely overlooked. This gap is partly because evaluating calibration itself is challenging in the PU setting—the absence of negative labels makes the standard ECE estimator inapplicable, which prevents practitioners from assessing the reliability of their PU classifiers.

In this paper, we address the problem of estimating the calibration error in PU learning settings. We introduce *PU-ECE*, a novel method for estimating the ECE using only PU data, without requiring negative labels. The core idea is to reframe the ECE computation by leveraging the statistical properties of PU datasets. By applying Bayes' rule within each bin, we can estimate from PU data the conditional expectation of the true label given the confidence score, which is necessary for the ECE computation. Our theoretical analysis establishes non-asymptotic bias bounds of PU-ECE, which guarantee the precision of our estimator from the TCE in a finite sample. With these bounds, we prove that with an optimal binning strategy, our estimator achieves a convergence rate that matches the convergence rate of the ECE in fully-supervised settings of $\mathcal{O}(n^{-1/3})$, where $n$ is the number of test samples (Futami & Fujisawa, 2024). An information-theoretic generalization error analysis of PU-ECE is also provided by formalizing the functional conditional mutual information (fCMI) (Steinke & Zakynthinou, 2020; Hellström & Durisi, 2022) for a PU setting. In addition, we demonstrate through extensive experiments on synthetic and benchmark datasets that our approach achieves estimation biases comparable to those of the fully-supervised method.

The rest of this paper is organized as follows. In Section 2, we formally describe the problem setting and define notations. In Section 3, we present the PU-ECE method and its theoretical properties, and in Section 4, we present its information-theoretic analysis. Section 5 provides experimental results to validate our approach. Finally, we conclude the paper in Section 6.

## 2 Problem Formulation

We denote random variables by capital letters and their realizations by lowercase letters. Let $\mathcal{X}$ be the input space and $\mathcal{Y} = \{0, 1\}$ be the label space. Let $P(X, Y)$ be the probability distribution over $\mathcal{X} \times \mathcal{Y}$, $P(X)$ be the marginal distribution of $X$, $P(X \mid Y = y)$ be the conditional distribution of $X$ given $Y = y \in \mathcal{Y}$, and $\pi_{\mathrm{P}} = P(Y = 1)$ be the class prior. The hypothesis space is denoted by $\mathcal{H} \subseteq \mathbb{R}^{\mathcal{X}}$ and a classifier $h \in \mathcal{H}$ outputs a real-valued score for a given input $x \in \mathcal{X}$. The model's confidence score is then obtained by applying an output activation function $\sigma : \mathbb{R} \to [0, 1]$, which we denote by $f(x) := \sigma(h(x))$. We assume $\sigma$ is measurable and monotone increasing. Examples include the sigmoid function and the probit function. If the sigmoid function $\sigma_{\mathrm{sig}}(z) := (1 + e^{-z})^{-1}$ is used as $\sigma$, we refer to $h(x)$ as the logit score. We define $\mathcal{F} = \{\sigma(h(\cdot)) \mid h \in \mathcal{H}\}$ for a fixed $\sigma$. We denote the parameterized classifier and its confidence score function by $h_w$ and $f_w$ with $w \in \mathcal{W}$, where $\mathcal{W}$ is the parameter space. If the parameter $w$ is fixed, we simply write $h$ and $f$. Let $\mathbb{1}_A$ be the indicator function, which returns 1 if the condition $A$ is true and 0 otherwise. We define $[n] = \{1, 2, \ldots, n\}$ for $n \in \mathbb{N}$. Let $I(X; Y \mid Z)$ denote the conditional mutual information between random variables $X$ and $Y$ given $Z$.

## 2.1 Supervised Learning

In supervised learning, the dataset is a set of independent and identically distributed (i.i.d.) samples drawn from $P(X, Y)$. Let $l : \mathbb{R} \times \mathcal{Y} \to \mathbb{R}_{\geq 0}$ be the loss function, e.g., the 0-1 loss (Wald, 1945) for the misclassification rate. Let $S_{\mathrm{PN}} := \left\{ \left( X_m^{\mathrm{PN}}, Y_m^{\mathrm{PN}} \right) \right\}_{m=1}^{n_{\mathrm{PN}}}$ be a dataset of $n_{\mathrm{PN}}$ samples drawn from $P(X, Y)$. The expected risk of $h$ on $P(X, Y)$ is given by $\mathbb{E}_{X,Y} [l(h(X), Y)]$. The empirical risk on $S_{\mathrm{PN}}$ is given by $\frac{1}{n_{\mathrm{PN}}} \sum_{m=1}^{n_{\mathrm{PN}}} l(h(X_m^{\mathrm{PN}}), Y_m^{\mathrm{PN}})$. A standard approach for training classifiers is to minimize this quantity, a principle known as Empirical Risk Minimization (ERM) (Vapnik & Chervonenkis, 1971).

## 2.2 Calibration Error Estimation

A classifier is said to be perfectly calibrated if its confidence score matches the true class probability (Murphy & Epstein, 1967; Gupta & Ramdas, 2021). Mathematically, this means that for all confidence scores $p \in [0, 1]$, we have $P(Y = 1 \mid f(X) = p) = p$ almost surely (Gupta & Ramdas, 2021). However, this ideal state is rarely achieved in practice (Guo et al., 2017). To quantify the deviation from this perfect calibration, the true calibration error (TCE) (Naeini et al., 2015; Gupta & Ramdas, 2021; Roelofs et al., 2022; Futami & Fujisawa, 2024) is used as a calibration metric.

### 2.2.1 True Calibration Error (TCE)

The TCE of $f$ on $P(X, Y)$ is defined as

$$\mathrm{TCE}(f) = \mathbb{E}_X \left[ |\mathbb{E}\left[ Y \mid f(X) \right] - f(X)| \right].$$

This measures the average absolute difference between the true conditional expectation of $Y$ given $f(X)$ and the confidence score $f(X)$. It quantifies how well the confidence scores align with the true probabilities of the positive class. A lower TCE indicates better calibration, meaning that the confidence scores are closer to the true probabilities. However, calculation of $\mathrm{TCE}(f)$ requires access to the conditional distribution $P(Y \mid f(X))$, which is usually unavailable.

### 2.2.2 Expected Calibration Error (ECE)

Instead, to estimate $\mathrm{TCE}(f)$, we often use a binning method (Zadrozny & Elkan, 2001; Guo et al., 2017; Futami & Fujisawa, 2024). This method involves a set of $B \in \mathbb{N}$ bins (intervals), denoted by $\mathcal{I} = \{I_b\}_{b=1}^{B}$, that covers the entire range of $f$. We define the true binned function as

$$f_{\mathcal{I}}(x) = \sum_{b=1}^{B} \mathbb{E}\left[ f(X) \mid f(X) \in I_b \right] \mathbb{1}_{f(x) \in I_b}.$$

Then its TCE is calculated as follows (Futami & Fujisawa, 2024):

$$\mathrm{TCE}(f_{\mathcal{I}}) := \sum_{b=1}^{B} P(f(X) \in I_b) \left| \mathbb{E}\left[ Y \mid f(X) \in I_b \right] - \mathbb{E}\left[ f(X) \mid f(X) \in I_b \right] \right|.$$

$\mathrm{TCE}(f_{\mathcal{I}})$ can be estimated from a supervised evaluation dataset $S_{\mathrm{e}} := \{(X_m^{\mathrm{e}}, Y_m^{\mathrm{e}})\}_{m=1}^{n_{\mathrm{e}}}$ of size $n_{\mathrm{e}} \in \mathbb{N}$, which can be a validation dataset or a test dataset, by the plug-in method. The resulting empirical estimate is called the expected calibration error (ECE):

$$\mathrm{ECE}(f, S_{\mathrm{e}}) := \sum_{b=1}^{B} p_b \left| \bar{y}_{b,S_{\mathrm{e}}} - \bar{f}_{b,S_{\mathrm{e}}} \right|, \tag{1}$$

where $p_b := \frac{\sum_{m=1}^{n_{\mathrm{e}}} \mathbb{1}_{f(X_m^{\mathrm{e}}) \in I_b}}{n_{\mathrm{e}}}$ is the fraction of samples in bin $b$, $\bar{y}_{b,S_{\mathrm{e}}} := \frac{\sum_{m=1}^{n_{\mathrm{e}}} \mathbb{1}_{f(X_m^{\mathrm{e}}) \in I_b} Y_m^{\mathrm{e}}}{\sum_{m=1}^{n_{\mathrm{e}}} \mathbb{1}_{f(X_m^{\mathrm{e}}) \in I_b}}$ is the average true label in bin $b$, and $\bar{f}_{b,S_{\mathrm{e}}} := \frac{\sum_{m=1}^{n_{\mathrm{e}}} \mathbb{1}_{f(X_m^{\mathrm{e}}) \in I_b} f(X_m^{\mathrm{e}})}{\sum_{m=1}^{n_{\mathrm{e}}} \mathbb{1}_{f(X_m^{\mathrm{e}}) \in I_b}}$ is the average confidence score in bin $b$. We assume $n_{\mathrm{e}} \geq 2B$, where $n_{\mathrm{e}}$ is the supervised evaluation dataset size and $B$ is the number of bins.

For $\mathcal{I}$, there are two common choices: uniform width binning (UWB) (Guo et al., 2017) and uniform mass binning (UMB) (Zadrozny & Elkan, 2001). In UWB, the bins are defined as $I_b = \left(\frac{b-1}{B}, \frac{b}{B}\right]$ for $b \in [B]$. In UMB, the bins are defined as $I_b = (u_{b-1}, u_b]$ for $b \in [B]$, where $u_0 = 0$, $u_b = f_{(k_b)}$ for $b \in [B-1]$, and $u_B = 1$. Here, $k_b = \lfloor \frac{n_e b}{B} \rfloor$, where $\lfloor \cdot \rfloor$ denotes the floor function that returns the greatest integer less than or equal to the input value, and $f_{(k)}$ denotes the $k$-th order statistic of the set $\{f(X_m^e)\}_{m=1}^{n_e}$.

## 2.3 Positive-Unlabeled Learning

Positive-unlabeled (PU) learning is a classification problem where only positive and unlabeled data are observed, but no negative data. Among various settings of PU learning (Elkan & Noto, 2008; Natarajan et al., 2013; du Plessis et al., 2014; Kato et al., 2019; Bekker et al., 2020), we focus on the selected-completely-at-random (SCAR) and case-control scenario. Concretely, we observe two independent sample sets: a set of unlabeled samples $S_{\mathrm{tr,U}} := \left\{X_m^{\mathrm{tr,U}}\right\}_{m=1}^{n_{\mathrm{tr,U}}} \overset{\text{i.i.d.}}{\sim} P(X)$ and a set of positive samples $S_{\mathrm{tr,P}} := \left\{X_m^{\mathrm{tr,P}}\right\}_{m=1}^{n_{\mathrm{tr,P}}} \overset{\text{i.i.d.}}{\sim} P(X \mid Y = 1)$, with the full training data denoted by $S_{\mathrm{tr}} = (S_{\mathrm{tr,U}}, S_{\mathrm{tr,P}})$ (du Plessis et al., 2014). This SCAR and case-control scenario is commonly used in the PU learning literature (Natarajan et al., 2013; du Plessis et al., 2014; Kiryo et al., 2017; Yao et al., 2021). Throughout the paper, we assume that the class prior $\pi_{\mathrm{P}} = P(Y = 1)$ is known—in practice, it can be estimated from data (Blanchard et al., 2010; Garg et al., 2021; Zhu et al., 2023).

In PU learning, an unbiased risk estimator of $h$ on $S_{\mathrm{tr}}$ is given as follows (Natarajan et al., 2013; du Plessis et al., 2014; 2015):

$$\mathcal{L}_{\mathrm{uPU}}(h) = \frac{\pi_{\mathrm{P}}}{n_{\mathrm{tr,P}}} \sum_{m=1}^{n_{\mathrm{tr,P}}} \left\{ l(h(X_m^{\mathrm{tr,P}}), 1) - l(h(X_m^{\mathrm{tr,P}}), 0) \right\} + \frac{1}{n_{\mathrm{tr,U}}} \sum_{m=1}^{n_{\mathrm{tr,U}}} l(h(X_m^{\mathrm{tr,U}}), 0). \tag{2}$$

The estimator is suitable for estimating the expected risk $\mathbb{E}\left[l(h(X), Y)\right]$ or training simple classifiers (du Plessis et al., 2014; 2015), but it does not work well for training more complex classifiers such as deep neural networks due to the *negative risk* issue (Kiryo et al., 2017). To overcome this issue, non-negative risk estimators were proposed (Kiryo et al., 2017; Lu et al., 2020). The non-negative risk estimator is given by

$$\mathcal{L}_{\mathrm{nnPU}}(h) = \frac{\pi_{\mathrm{P}}}{n_{\mathrm{tr,P}}} \sum_{m=1}^{n_{\mathrm{tr,P}}} l(h(X_m^{\mathrm{tr,P}}), 1) + l_{\mathrm{cc}}\left( \frac{1}{n_{\mathrm{tr,U}}} \sum_{m=1}^{n_{\mathrm{tr,U}}} l(h(X_m^{\mathrm{tr,U}}), 0) - \frac{\pi_{\mathrm{P}}}{n_{\mathrm{tr,P}}} \sum_{m=1}^{n_{\mathrm{tr,P}}} l(h(X_m^{\mathrm{tr,P}}), 0) \right),$$

where $l_{\mathrm{cc}} : \mathbb{R} \to \mathbb{R}_{\geq 0}$ is called a consistent-correction function. Several functions have been proposed for use as $l_{\mathrm{cc}}$, e.g., $\max\{x, 0\}$ (Kiryo et al., 2017; Lu et al., 2020).

## 3 PU-ECE: An Estimator for Calibration Error in PU Learning

Conventional ECE estimation, as shown in Eq. (1), requires labeled data (both positive and negative), but in the PU environment, negative data are unavailable, as introduced in Section 2.3. The main challenges are (i) estimating the true positive rate within bins, and (ii) handling the fact that the data do not come from a single distribution $P(X, Y)$ but from $P(X)$ and $P(X|Y = 1)$. In this section, we propose a method to estimate the ECE in the PU environment. The proposed method is based on the idea of using positive and unlabeled data to estimate $\mathbb{E}[Y \mid f(X) \in I_b]$. We denote the PU evaluation data as $S_{\mathrm{U}} := \{X_m^{\mathrm{U}}\}_{m=1}^{n_{\mathrm{U}}}$, $S_{\mathrm{P}} := \{X_m^{\mathrm{P}}\}_{m=1}^{n_{\mathrm{P}}}$, and $S_{\mathrm{PU}} := (S_{\mathrm{U}}, S_{\mathrm{P}})$. The evaluation data are independent of the training data and $f$, and can be a validation dataset or a test dataset in practice.

### 3.1 Proposed Estimator

Since $Y \in \{0, 1\}$, the conditional expectation can be expressed as

$$\mathbb{E}[Y \mid f(X) \in I_b] = P(Y = 1 \mid f(X) \in I_b) = \frac{P(Y = 1, f(X) \in I_b)}{P(f(X) \in I_b)} = \frac{\pi_{\mathrm{P}} P(f(X) \in I_b \mid Y = 1)}{P(f(X) \in I_b)}. \tag{3}$$

In the PU learning setting, we have access to positive samples $S_{\mathrm{P}} = \{X_m^{\mathrm{P}}\}_{m=1}^{n_{\mathrm{P}}}$ drawn i.i.d. from $P(X|Y = 1)$ and unlabeled samples $S_{\mathrm{U}} = \{X_m^{\mathrm{U}}\}_{m=1}^{n_{\mathrm{U}}}$ drawn i.i.d. from $P(X)$. The term $P(f(X) \in I_b \mid Y = 1)$ in Eq. (3) can be estimated by the empirical frequency of positive samples falling into bin $I_b$: $p_{b,\mathrm{P}} := \frac{1}{n_{\mathrm{P}}} \sum_{m=1}^{n_{\mathrm{P}}} \mathbb{1}_{f(X_m^{\mathrm{P}}) \in I_b}$. Similarly, $P(f(X) \in I_b)$ can be estimated by the empirical frequency of unlabeled samples falling into bin $I_b$: $p_{b,\mathrm{U}} := \frac{1}{n_{\mathrm{U}}} \sum_{m=1}^{n_{\mathrm{U}}} \mathbb{1}_{f(X_m^{\mathrm{U}}) \in I_b}$. These estimators are unbiased, that is, $\mathbb{E}[p_{b,\mathrm{P}}] = P(f(X) \in I_b \mid Y = 1)$ and $\mathbb{E}[p_{b,\mathrm{U}}] = P(f(X) \in I_b)$. Let $|I_{b,\mathrm{P}}| = \sum_{m=1}^{n_{\mathrm{P}}} \mathbb{1}_{f(X_m^{\mathrm{P}}) \in I_b}$ and $|I_{b,\mathrm{U}}| = \sum_{m=1}^{n_{\mathrm{U}}} \mathbb{1}_{f(X_m^{\mathrm{U}}) \in I_b}$ be the number of positive and unlabeled samples in bin $I_b$, respectively. Then $p_{b,\mathrm{P}} = |I_{b,\mathrm{P}}|/n_{\mathrm{P}}$ and $p_{b,\mathrm{U}} = |I_{b,\mathrm{U}}|/n_{\mathrm{U}}$. Using these estimates, an estimator for the conditional probability $P(Y = 1 \mid f(X) \in I_b)$ is naively obtained as $\hat{y}_{b,S_{\mathrm{PU}}} := \frac{\pi_{\mathrm{P}} p_{b,\mathrm{P}}}{p_{b,\mathrm{U}}}$. The ECE for PU learning (PU-ECE) can then be estimated as

$$\sum_{b=1}^{B} p_{b,\mathrm{U}} \left| \hat{y}_{b,S_{\mathrm{PU}}} - \bar{f}_{b,S_{\mathrm{PU}}} \right|,$$

where $\bar{f}_{b,S_{\mathrm{PU}}} := \frac{\sum_{m=1}^{n_{\mathrm{U}}} \mathbb{1}_{f(X_m^{\mathrm{U}}) \in I_b} f(X_m^{\mathrm{U}})}{|I_{b,\mathrm{U}}|}$ is the average confidence score for unlabeled samples in bin $I_b$. If $|I_{b,\mathrm{U}}| = 0$, then $p_{b,\mathrm{U}} = 0$. In this case, $\hat{y}_{b,S_{\mathrm{PU}}}$ would involve division by zero. However, this is a removable singularity. For all cases where $p_{b,\mathrm{U}} > 0$, we can algebraically reformulate the estimator as follows:

$$\mathrm{ECE}_{\mathrm{PU}}(f, S_{\mathrm{PU}}) = \sum_{b=1}^{B} \left| \frac{\pi_{\mathrm{P}}}{n_{\mathrm{P}}} \sum_{m=1}^{n_{\mathrm{P}}} \mathbb{1}_{f(X_m^{\mathrm{P}}) \in I_b} - \frac{1}{n_{\mathrm{U}}} \sum_{m=1}^{n_{\mathrm{U}}} \mathbb{1}_{f(X_m^{\mathrm{U}}) \in I_b} f(X_m^{\mathrm{U}}) \right|.$$

This new form is mathematically equivalent for all $p_{b,\mathrm{U}} > 0$, and remains well-defined at $p_{b,\mathrm{U}} = 0$, and its value at $p_{b,\mathrm{U}} = 0$ coincides with the limit of the estimator as $p_{b,\mathrm{U}} \to 0$. The computational complexity of PU-ECE is $\mathcal{O}(n_{\mathrm{U}} + n_{\mathrm{P}})$ given the bins. Including bin construction (sorting), the total complexity is $\mathcal{O}(n_{\mathrm{U}} \log n_{\mathrm{U}} + n_{\mathrm{P}} \log n_{\mathrm{P}})$, which is the same as that of the standard ECE.

We define UWB and UMB for the PU setting as follows: in UWB, the bins are defined in the same way as in supervised learning, while in UMB, the bins are defined by *the unlabeled data*; $I_b = (u_{b-1}, u_b]$ for $b \in [B]$, where $u_0 = 0$, $u_b = f_{(k_b)}$ for $b \in [B - 1]$, and $u_B = 1$. Here, $k_b = \lfloor \frac{n_{\mathrm{U}} b}{B} \rfloor$, and $f_{(k)}$ denotes the $k$-th order statistic of the set $\{f(X_m^{\mathrm{U}})\}_{m=1}^{n_{\mathrm{U}}}$, i.e., the $k$-th smallest value.

## 3.2 Bias Analysis of PU-ECE

We analyze the bias of the proposed estimator given the evaluation data. In the analysis, we assume the following:

**Assumption 1.** *The number of positive samples $n_{\mathrm{P}}$ satisfies $n_{\mathrm{P}} \geq 1$. The number of unlabeled samples $n_{\mathrm{U}}$ satisfies $n_{\mathrm{U}} \geq 2B$ for UMB and $n_{\mathrm{U}} \geq 1$ for UWB.*

**Assumption 2.** *$f(X)$ is absolutely continuous with respect to the Lebesgue measure, i.e., $f(X)$ has a probability density function.*

**Assumption 3.** *$\mathbb{E}[Y \mid f(X)]$ satisfies L-Lipschitz continuity with respect to $f(X)$, i.e., $|\mathbb{E}[Y \mid f(X) = v] - \mathbb{E}[Y \mid f(X) = v']| \leq L|v - v'|$ for all $v, v' \in [0, 1]$.*

The (total) bias of the PU-ECE estimator is defined as

$$\mathrm{Bias}_{\mathrm{tot}}(f, S_{\mathrm{PU}}) := |\mathrm{ECE}_{\mathrm{PU}}(f, S_{\mathrm{PU}}) - \mathrm{TCE}(f)|.$$

Then, we have the following theorem, which provides non-asymptotic bias bounds for the PU-ECE estimator.

**Theorem 1.** *Under Assumptions 1, 2 and 3, we have*

$$\mathbb{E}_{S_{\mathrm{PU}}}[\mathrm{Bias}_{\mathrm{tot}}(f, S_{\mathrm{PU}})] \leq \begin{cases} \frac{1+L}{B} + \sqrt{2\left(\frac{\pi_{\mathrm{P}}^2}{n_{\mathrm{P}}} + \frac{1}{n_{\mathrm{U}}}\right) B \log 2} & \text{(for UWB)}, \\[3mm] \frac{1+L}{B} + \frac{(3+2L)B}{n_{\mathrm{U}}-B} + \sqrt{2B \log 2}\left(\frac{1+L}{\sqrt{n_{\mathrm{U}}-B}} + \sqrt{\frac{\pi_{\mathrm{P}}^2}{n_{\mathrm{P}}} + \frac{1}{n_{\mathrm{U}}-B}}\right) & \text{(for UMB)}. \end{cases}$$

*Furthermore, for any $\delta \in (0,1)$, with probability at least $1 - \delta$ over $S_{\mathrm{PU}}$, we have*

$$
\mathrm{Bias}_{\mathrm{tot}}(f, S_{\mathrm{PU}}) \leq \begin{cases} \frac{1+L}{B} + \sqrt{2\left(\frac{\pi_{\mathrm{P}}^2}{n_{\mathrm{P}}} + \frac{1}{n_{\mathrm{U}}}\right)(B \log 2 + \log(1/\delta))} & \text{(for UWB),} \\[2ex] \frac{1+L}{B} + \frac{(3+2L)B}{n_{\mathrm{U}}-B} + \sqrt{2(B \log 2 + \log(1/\delta))}\left(\frac{1+L}{\sqrt{n_{\mathrm{U}}-B}} + \sqrt{\frac{\pi_{\mathrm{P}}^2}{n_{\mathrm{P}}} + \frac{1}{n_{\mathrm{U}}-B}}\right) & \text{(for UMB).} \end{cases}
$$

The proof of Theorem 1 is provided in Appendix A.1. Briefly, the proof is based on decomposing the total bias into two parts: the bias due to binning and the bias due to the estimation by the empirical frequency, and then bounding each part separately by concentration inequalities.

Minimizing the upper bound of the total bias with respect to $B$, we have the following upper bound orders:

**Corollary 1.** *Under Assumption 1, the scale of the optimal $B$ that minimizes the upper bound in Theorem 1 for both UWB and UMB is given by*

$$
B = \Theta\left(\left(\frac{\pi_{\mathrm{P}}^2}{n_{\mathrm{P}}} + \frac{1}{n_{\mathrm{U}}}\right)^{-\frac{1}{3}}\right),
$$

*and the corresponding expected total bias convergence rate is given by*

$$
\mathbb{E}_{S_{\mathrm{PU}}}[\mathrm{Bias}_{\mathrm{tot}}(f, S_{\mathrm{PU}})] = \mathcal{O}\left(\left(\frac{\pi_{\mathrm{P}}^2}{n_{\mathrm{P}}} + \frac{1}{n_{\mathrm{U}}}\right)^{\frac{1}{3}}\right).
$$

The proof can be found in Appendix A.2. In practice, this means that to minimize the bias of the PU-ECE estimator, the number of bins $B$ should be chosen according to the formula above, balancing the sample sizes of the positive and unlabeled data. The result corresponds to the convergence rate of the supervised setting; the optimal $B$ is $\mathcal{O}(n^{\frac{1}{3}})$ and the expected total bias is $\mathcal{O}(n^{-\frac{1}{3}})$, where $n$ is the size of the fully-labeled evaluation data (Futami & Fujisawa, 2024). The term $\pi_{\mathrm{P}}^2$ in these expressions is a constant factor that does not depend on the sample size, but it indicates that a smaller class prior $\pi_{\mathrm{P}}$ leads to a smaller expected bias.

In practical PU learning scenarios, the true class prior $\pi_{\mathrm{P}}$ is often unknown and must be estimated from data (Bepler et al., 2019; Ito & Sugiyama, 2023), which introduces additional estimation error into the calibration process. It is therefore important to understand how inaccuracies in class prior estimation affect PU-ECE. We also analyze how the class prior estimation error affects the PU-ECE estimation.

**Theorem 2** (Error bound due to class prior estimation error). *Let $\hat{\pi}_{\mathrm{P}}$ be an estimate of the class prior $\pi_{\mathrm{P}}$, and define the PU-ECE estimator using $\hat{\pi}_{\mathrm{P}}$ as*

$$
\widehat{\mathrm{ECE}}_{\mathrm{PU}}(f, S_{\mathrm{PU}}) = \sum_{b=1}^{B}\left|\frac{\hat{\pi}_{\mathrm{P}}}{n_{\mathrm{P}}}\sum_{m=1}^{n_{\mathrm{P}}} \mathbb{1}_{f(X_m^{\mathrm{P}}) \in I_b} - \frac{1}{n_{\mathrm{U}}}\sum_{m=1}^{n_{\mathrm{U}}} f(X_m^{\mathrm{U}})\mathbb{1}_{f(X_m^{\mathrm{U}}) \in I_b}\right|.
$$

*Then, we have the following bound on the difference between the PU-ECE with the true prior and that with the estimated prior:*

$$
\left|\mathrm{ECE}_{\mathrm{PU}}(f, S_{\mathrm{PU}}) - \widehat{\mathrm{ECE}}_{\mathrm{PU}}(f, S_{\mathrm{PU}})\right| \leq |\pi_{\mathrm{P}} - \hat{\pi}_{\mathrm{P}}|.
$$

The proof is provided in Appendix A.3. This inequality does not require expectation and holds for any $f$ and $S_{\mathrm{PU}}$. This theorem shows that when the class prior is unknown and must be estimated, the calibration error estimation incurs an additional error of at most the class prior estimation error.

## 4 Information-Theoretic Generalization Analysis of PU-ECE

The previous analysis focused on the bias of the ECE estimation using evaluation data, which is independent of the training data used for learning $f_w$. However, in practice, we may recalibrate classifiers using a recalibration dataset and then evaluate the PU-ECE on the same recalibration dataset (Kumar et al., 2019),

or we may calibrate classifiers using the training dataset (Futami & Fujisawa, 2024). In these cases, the data used for PU-ECE evaluation and the classifier recalibration/training are no longer independent. Therefore, analyzing the generalization error of the PU-ECE is crucial to ensure reliable model selection and evaluation in practice. In this section, we extend the conditional mutual information (CMI) based analysis framework (Steinke & Zakynthinou, 2020; Haghifam et al., 2021) used in supervised learning to the PU setting and analyze the generalization error of PU-ECE using it.

## 4.1 Information-Theoretic Generalization Error Analysis for Supervised Learning

A rigorous understanding of the generalization error is crucial for evaluating the reliability of machine learning algorithms. Classical paradigms for assessing generalization include the concepts of Vapnik-Chervonenkis (VC) dimension (Vapnik & Chervonenkis, 1968; Blumer et al., 1989) and metric entropy (Kolmogorov & Tikhomirov, 1959; Dudley, 1974). The VC dimension quantifies how many points a hypothesis class can shatter, while metric entropy measures how finely a function class can be approximated using a finite set of representative functions at a given precision. Despite the established significance of these tools, recent advances in information-theoretic perspectives, particularly those centered on CMI (Steinke & Zakynthinou, 2020; Haghifam et al., 2021), have provided new insights into generalization. For example, functional conditional mutual information (fCMI) (Steinke & Zakynthinou, 2020) measures the dependency between the predictions of a learned model on a supersample (the collection of both training and held-out data) and the random split that determines which part of the supersample is used for training, conditioned on the supersample itself. By bounding this data-model dependency, CMI-based analysis methods provide new insights into why certain learning algorithms exhibit strong generalization performance. For instance, Futami & Fujisawa (2024) applied this approach to derive non-asymptotic generalization error bounds for the supervised ECE in the supervised setting.

## 4.2 Functional Conditional Mutual Information (fCMI) for PU Learning

To analyze the generalization error, we extend the CMI-based analysis framework to the PU setting. We consider a supersample $\tilde{X} = [\tilde{X}^{\mathrm{U}}; \tilde{X}^{\mathrm{P}}] \in \mathcal{X}^{(n_{\mathrm{tr,U}} + n_{\mathrm{tr,P}}) \times 2}$, which is a $(n_{\mathrm{tr,U}} + n_{\mathrm{tr,P}}) \times 2$ array of samples. [2] Let $\tilde{X}^{\mathrm{U}} \in \mathcal{X}^{n_{\mathrm{tr,U}} \times 2}$ be the unlabeled supersample and $\tilde{X}^{\mathrm{P}} \in \mathcal{X}^{n_{\mathrm{tr,P}} \times 2}$ be the positive supersample. For convenience, we use the 0-start-index for the columns of the arrays and the 1-start-index for the rows of the arrays. We denote the $i$-th row of $\tilde{X}$ by $\tilde{X}_i$. Analogous to $\tilde{X}$, we have a supersample membership vector $M := [M^{\mathrm{U}}; M^{\mathrm{P}}] \in \{0, 1\}^{n_{\mathrm{tr,U}} + n_{\mathrm{tr,P}}}$, where $M^{\mathrm{U}} \in \{0, 1\}^{n_{\mathrm{tr,U}}}$ and $M^{\mathrm{P}} \in \{0, 1\}^{n_{\mathrm{tr,P}}}$ are the membership vectors for the unlabeled and positive supersamples, respectively. The entry $M_i$ specifies which column of the $i$-th row $\tilde{X}_i$ is retained for the $i$-th data point for training: $M_i = 0$ selects the left column (index 0) and $M_i = 1$ selects the right column (index 1). All entries are drawn independently as $M_i \sim \mathrm{Bern}(1/2)$. We denote the PU training vector by $\tilde{X}_M$. The remaining entries of $\tilde{X}$ that are not selected by $M$ can be regarded as the PU evaluation data. Let $\mathcal{A} : \mathcal{X}^{n_{\mathrm{tr,U}} + n_{\mathrm{tr,P}}} \times \mathcal{R} \to \mathcal{W}$ be a randomized algorithm that maps a training dataset and a source of randomness to the parameter space $\mathcal{W}$. The source of randomness, denoted by $R \in \mathcal{R}$, is independent of any other random variables. Given a training dataset $S_{\mathrm{tr}}$ and randomness $R$, the algorithm outputs the parameters $W = \mathcal{A}(S_{\mathrm{tr}}, R) \in \mathcal{W}$, and we denote the corresponding predictor by $f_W$.

We define the functional conditional mutual information (fCMI) for PU data as

$$\mathrm{fCMI} := I(f_W(\tilde{X}); M | \tilde{X}),$$

where $f_W(\tilde{X})$ is the matrix calculated by the elementwise application of $f_W$ to $\tilde{X}$. Similarly, we also define fCMI on $\tilde{X}^{\mathrm{U}}$ as

$$\mathrm{fCMI}_{\mathrm{U}} := I(f_W(\tilde{X}^{\mathrm{U}}); M | \tilde{X}).$$

By the data processing inequality, we have $\mathrm{fCMI}_{\mathrm{U}} \leq \mathrm{fCMI}$. The PU fCMI metrics can be bounded by model complexity measures such as the VC dimension (Steinke & Zakynthinou, 2020; Harutyunyan et al., 2021) or the metric entropy (Futami & Fujisawa, 2024). Detailed discussions and definitions of these quantities are provided in Appendix B.

---

[2] We use the term "array" for notational convenience, although each entry is an element of $\mathcal{X}$, which can be a high-dimensional vector rather than a scalar.

### 4.3 Generalization Bound for PU-ECE

This subsection analyzes the generalization error of PU-ECE using the fCMIs defined above. We follow Futami & Fujisawa (2024) and use $\text{Bias}_{\text{tot}}(f_W, S_{\text{tr}}) := |\text{TCE}(f_W) - \text{ECE}_{\text{PU}}(f_W, S_{\text{tr}})|$ as the generalization error of $\text{ECE}_{\text{PU}}(f_W, S_{\text{tr}})$.

**Theorem 3** (Expected generalization error bound of PU-ECE)**.** *Under Assumptions 2 and 3, we have*

$$\mathbb{E}_{R,S_{\text{tr}}}\left[\text{Bias}_{\text{tot}}(f_W, S_{\text{tr}})\right]$$

$$\leq \begin{cases} \frac{1+L}{B} + \sqrt{8\left(\frac{\pi_{\text{P}}^2}{n_{\text{tr,P}}} + \frac{1}{n_{\text{tr,U}}}\right)\{\text{fCMI} + B\log 2\}} & \text{(for UWB)}, \\ \frac{1+L}{B} + \sqrt{8\left(\frac{\pi_{\text{P}}^2}{n_{\text{tr,P}}} + \frac{1}{n_{\text{tr,U}}}\right)\{\text{fCMI} + B\log 2\}} + (1+L)\sqrt{\frac{8}{n_{\text{tr,U}}}\{\text{fCMI}_{\text{U}} + B\log 2\}} & \text{(for UMB)}. \end{cases}$$

The proof of Theorem 3 is provided in Appendix C.1. Briefly, the proof is similar to the proof of Theorem 1, except the dependency on the training data is handled by the fCMIs.

Similar to Corollary 1, the optimal bin size to minimize the upper bound in Theorem 3 and the order of the upper bound can also be derived if a learning algorithm has a sufficiently small fCMI. For instance, such an assumption holds when $\mathcal{H}$ is a finite VC class since $\mathcal{F}$ is also a finite VC class, and hence fCMI is bounded by $\mathcal{O}(\log(n_{\text{tr,P}} + n_{\text{tr,U}}))$ (Steinke & Zakynthinou, 2020). For such a case, we can derive the following result.

**Corollary 2** (Total bias upper bound order)**.** *Under Assumptions 2 and 3 and assuming* $\text{fCMI} = \mathcal{O}\left(\left(\frac{\pi_{\text{P}}^2}{n_{\text{tr,P}}} + \frac{1}{n_{\text{tr,U}}}\right)^{-1/3}\right)$*, the scale of the optimal $B$ that minimizes the upper bound in Theorem 3 is given by*

$$B = \Theta\left(\left(\frac{\pi_{\text{P}}^2}{n_{\text{tr,P}}} + \frac{1}{n_{\text{tr,U}}}\right)^{-1/3}\right),$$

*and the convergence rate of the corresponding expected generalization error is given by*

$$\mathbb{E}_{R,S_{\text{tr}}}\left[\text{Bias}_{\text{tot}}(f_W, S_{\text{tr}})\right] = \mathcal{O}\left(\left(\frac{\pi_{\text{P}}^2}{n_{\text{tr,P}}} + \frac{1}{n_{\text{tr,U}}}\right)^{1/3}\right).$$

The proof is provided in Appendix A.2.

By Corollaries 1 and 2, $\mathbb{E}_{R,S_{\text{tr}}}\left[\text{Bias}_{\text{tot}}(f_W, S_{\text{tr}})\right] \ll \mathbb{E}_{S_{\text{PU}}}\left[\text{Bias}_{\text{tot}}(f_W, S_{\text{PU}})\right]$ if $n_{\text{tr,P}} \gg n_{\text{P}}$ and $n_{\text{tr,U}} \gg n_{\text{U}}$ and the classifier generalizes well. This would actually happen in practice since the training dataset is often much larger than the evaluation dataset. Such a property is also observed in the supervised setting (Futami & Fujisawa, 2024). This represents an important future direction for PU learning to develop a method that jointly optimizes the TCE and accuracy, although this is beyond the scope of this paper.

## 5 Experiments

In this section, we empirically evaluate the theoretical properties of the proposed PU-ECE estimator. We empirically validate the theoretical bias and convergence properties of our proposed PU-ECE estimator on both synthetic and real-world benchmark datasets.

### 5.1 Experimental Setup

The sigmoid function was used as the output activation function. Results are averaged over 100 trials, each with a different random seed. The code for all experiments was implemented in Python and is available at `https://openreview.net/forum?id=SvoBtLIrP`.

| Dataset | $n_{\text{tr,U}}$ | $\pi_{\text{P}}$ | Model | Learning rate |
|---|---|---|---|---|
| MNIST (LeCun et al., 1998) | 60,000 | 0.49 | 2-layer MLP (784-100-100-1) | $1.0 \times 10^{-3}$ |
| CIFAR-10 (Krizhevsky, 2009) | 50,000 | 0.40 | ResNet-18 (He et al., 2016) | $1.0 \times 10^{-5}$ |
| DDI (Herrero-Zazo et al., 2013) | 27,792 | 0.14 | R-BERT (Wu & He, 2019) | $2.0 \times 10^{-5}$ |

Table 1: Specification of benchmark datasets, models, and learning rates. $n_{\text{tr,U}}$ denotes the number of unlabeled samples for training PU classifiers.

**Synthetic data**  Following previous work (Vaicenavicius et al., 2019; Zhang et al., 2020; Futami & Fujisawa, 2024), we generated data from a logistic model where $P(Y = 1 \mid X = x) = \sigma_{\text{sig}}(2x)$, with the conditional probability densities $p(X) = 0.5\,\mathcal{N}(x; 1, 1) + 0.5\,\mathcal{N}(x; -1, 1)$. We evaluated a classifier $f(x) = \sigma_{\text{sig}}(\beta_0 + \beta_1 x)$ under two settings: a less-calibrated case ($\beta_0 = -0.5, \beta_1 = 1.5$) and a better-calibrated case ($\beta_0 = -0.2, \beta_1 = 1.9$). The ground-truth TCE($f$) was computed by the trapezoidal rule. Further details are provided in Appendix D.

**Benchmark data**  We used the MNIST, CIFAR-10 and DDI datasets, with specifications detailed in Table 1. Since these datasets are multiclass, we converted them into binary classification tasks by selecting positive classes and treating all other classes as negative. For MNIST, we used the digits 5-9 as the positive class, while for CIFAR-10, we selected the vehicle class (airplane, automobile, ship, and truck) as the positive class. For DDI, we treated drug pairs with an annotated drug-drug interaction (DDI) relation as the positive class, and all other pairs (without a DDI relation) as the negative class. Classifiers were trained using the nnPU learning method (Kiryo et al., 2017) for MNIST and CIFAR-10 and the imbalanced nnPU learning method (Su et al., 2021) for DDI, and the outputs of the classifiers were used as logit scores. The logistic loss (a.k.a. binary cross-entropy loss) $\log(\sigma_{\text{sig}}((2y - 1)h(x)))$ or sigmoid loss $\sigma_{\text{sig}}(-(2y - 1)h(x))$ was used as the loss function, depending on the experiment. The sigmoid loss was used as a noise-robust (Ghosh et al., 2017) and classification-calibrated (Charoenphakdee et al., 2019) loss function, and hence are widely used in weakly-supervised learning including PU learning (Kiryo et al., 2017; Chen et al., 2020; Zhao et al., 2022; Ye et al., 2023). Although the sigmoid loss is not strictly proper (Charoenphakdee et al., 2019), its confidence scores are conventionally interpreted as class-posterior probabilities (Platt, 1999; Chen et al., 2020; Ye et al., 2023). The sigmoid loss was used for MNIST and CIFAR-10, while the logistic loss was used for DDI. Classifiers were trained on PU training data with 10,000 positive for MNIST and CIFAR-10 and 2,000 positive for DDI, and $n_{\text{tr,U}}$ unlabeled samples specified in Table 1. The other details are provided in Appendix F.

**Ground-truth TCE computation**  To compute TCE($f$) for these datasets, we first need to assume parametric models for the true conditional probability densities. Fortunately, the TCE can be computed using the distribution of the logit score $h(X)$ or the confidence score $f(X)$, without modeling the distribution of $X$, which would otherwise suffer from the curse of dimensionality (see Appendix E for details). We observed that the confidence scores were heavily saturated near 0 or 1, while the logit scores were spread out. Therefore, we modeled the logit score $h(X)$ instead of the confidence score $f(X)$, specifically modeling the distributions of the logit score, $P(h(X) \mid Y = 1)$ and $P(h(X) \mid Y = 0)$. The empirical logit histograms were multi-modal, so a single-mode parametric family, such as the logistic-beta distribution used by Roelofs et al. (2022), is inadequate. For instance, the MNIST logits exhibited two clear peaks. Instead, we fitted Gaussian Mixture Models (GMMs) to $P(h(X) \mid Y = 1)$ and $P(h(X) \mid Y = 0)$, selecting the number of components for each GMM from 1 to 10 via the Bayesian Information Criterion (BIC) (Schwarz, 1978). In MNIST, the GMMs with 4 positive class components and 3 negative class components were selected, while in CIFAR-10, the GMMs with 2 positive class components and 2 negative class components were selected, and in DDI, the GMMs with 3 positive class components and 3 negative class components were selected. With these estimated densities, we sampled the logit scores from the GMMs. TCE($f$) was calculated using the trapezoidal rule at $10,000$ points.

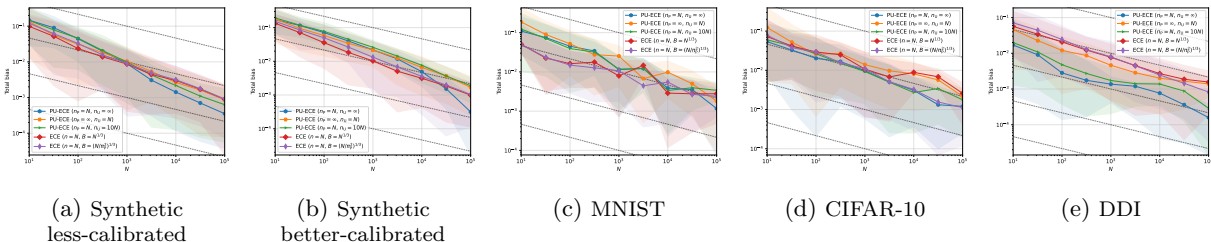

| (a) Synthetic less-calibrated | (b) Synthetic better-calibrated | (c) MNIST | (d) CIFAR-10 | (e) DDI |

Figure 1: Total bias of ECE and PU-ECE, as a function of sample size $N$. Both axes are log-scale. The lines represent the total bias, and the shaded areas represent the 90% percentile interval. The dashed lines represent the theoretical convergence rate of $\mathcal{O}(N^{-1/3})$.

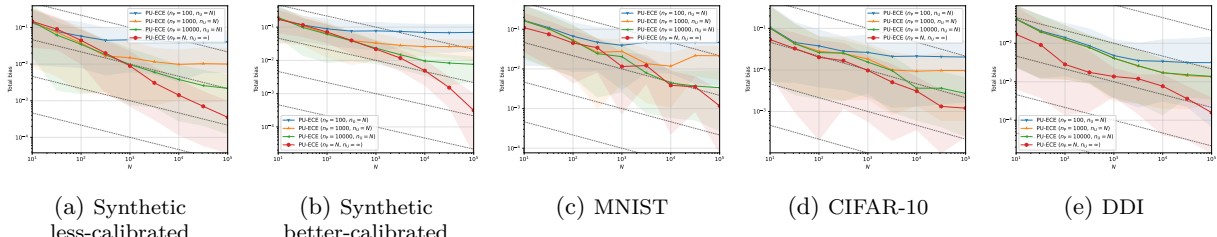

| (a) Synthetic less-calibrated | (b) Synthetic better-calibrated | (c) MNIST | (d) CIFAR-10 | (e) DDI |

Figure 2: Convergence of total bias of PU-ECE varying $n_{\mathrm{P}}$. Both axes are log-scale. The lines represent the total bias, and the shaded areas represent the 90% percentile interval. The dashed lines represent the theoretical convergence rate of $\mathcal{O}(N^{-1/3})$.

**Binning strategy** We use UMB by default, but we also evaluate UWB side-by-side to study stability in Appendix I. The number of bins $B$ was set to $\lceil(\pi_{\mathrm{P}}^2/n_{\mathrm{P}}+1/n_{\mathrm{U}})^{-1/3}\rceil$ for PU-ECE and $\lceil n_{\mathrm{PN}}^{1/3}\rceil$ or $\lceil(n_{\mathrm{PN}}/\pi_{\mathrm{P}}^2)^{1/3}\rceil$ for ECE, following the optimal order derived in our analysis and in Futami & Fujisawa (2024).

## 5.2 Empirical Bias Convergence Rate

Let $N$ denote an integer ranging from 10 to 10,000. We evaluated the convergence behavior by varying $N$, comparing PU-ECE in settings with $(n_{\mathrm{P}} = N, n_{\mathrm{U}} = \infty)$, $(n_{\mathrm{P}} = \infty, n_{\mathrm{U}} = N)$, $(n_{\mathrm{P}} = N, n_{\mathrm{U}} = 10N)$, and standard ECE with $n = N$. For settings with either $n_{\mathrm{U}} = \infty$ or $n_{\mathrm{P}} = \infty$, the corresponding components (e.g., $P(f(X) \in I_b)$ when $n_{\mathrm{U}} = \infty$) were computed via numerical integration using the trapezoidal rule with up to 10,000 points.

Figure 1 shows the total bias of PU-ECE and standard ECE (absolute bias compared to the TCE) on both synthetic and benchmark datasets. This experiment is designed to validate the theoretical convergence rate of our estimator. According to our analysis, the total bias of PU-ECE is expected to decrease at a rate of $\mathcal{O}((\pi_{\mathrm{P}}^2/n_{\mathrm{P}}+1/n_{\mathrm{U}})^{-1/3})$. When both $n_{\mathrm{P}}$ and $n_{\mathrm{U}}$ grow proportionally with $N$ (or are set to $\infty$), this simplifies to $\mathcal{O}(N^{-1/3})$.

The empirical results clearly exhibit this $\mathcal{O}(N^{-1/3})$ convergence, providing strong evidence in support of our theoretical claims. Notably, this rate matches the known optimal convergence rate for standard ECE (Futami & Fujisawa, 2024), demonstrating that PU-ECE can achieve competitive calibration performance even without access to fully labeled data.

### 5.2.1 Finite Positive Sample Size Effect

We also investigated the effect of fixing $n_{\mathrm{P}}$ by comparing PU-ECE with $(n_{\mathrm{P}} = 100, n_{\mathrm{U}} = N)$, $(n_{\mathrm{P}} = 1,000, n_{\mathrm{U}} = N)$, $(n_{\mathrm{P}} = 10,000, n_{\mathrm{U}} = N)$, and $(n_{\mathrm{P}} = N, n_{\mathrm{U}} = \infty)$.

Figure 2 investigates the effect of fixing $n_{\mathrm{P}}$ while increasing $n_{\mathrm{U}}$. We observe that PU-ECE converges to the limiting case of PU-ECE with $n_{\mathrm{P}} = N, n_{\mathrm{U}} = \infty$ but not to the true TCE. This behavior is consistent with

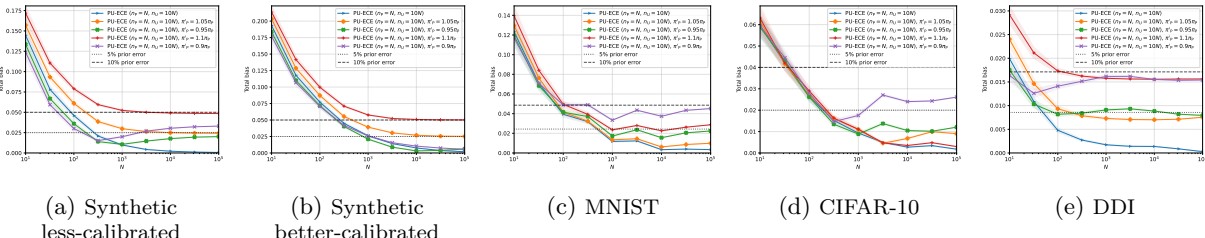

| (a) Synthetic less-calibrated | (b) Synthetic better-calibrated | (c) MNIST | (d) CIFAR-10 | (e) DDI |

Figure 3: Effect of prior estimation error on PU-ECE. The lines represent the total bias, and the shaded areas represent the standard error. We perturb the class prior $\pi_P$ by $\pm 5\%$ and $\pm 10\%$ in PU-ECE computation and observe shifts proportional to the prior estimation error, consistent with Theorem 2.

our theoretical prediction: even as $n_U \to \infty$, the bias remains bounded below by the term depending on $n_P$. This residual error creates an error floor that prevents convergence to the TCE when $n_P$ is finite.

These results highlight a key practical insight: sufficiently large sets of both positive and unlabeled samples are essential for effectively estimating the TCE.

### 5.3 Robustness to Class Prior Estimation Error

In many PU applications, the class prior $\pi_P$ is estimated from data. Section 3.2 (Theorem 2) shows that using an estimated prior $\hat{\pi}_P$ perturbs PU-ECE by at most $|\pi_P - \hat{\pi}_P|$. Here, $\pm 5\%$ and $\pm 10\%$ prior estimation errors mean $|\pi_P - \hat{\pi}_P| = 0.05\,\pi_P$ and $0.10\,\pi_P$, respectively. Since $\pi_P$ is known in our experiments, we can compute the corresponding error bound from Theorem 2 and compare it with the observed total bias. Note that this bound controls only the perturbation due to prior misspecification and does not bound the overall total bias. Consequently, at small sample sizes the observed total bias can exceed the prior-error bound. Figure 3 shows that as the data size grows and the total bias itself becomes small, it no longer exceeds the bound and typically converges to a value below it. The bound becomes tight only in the synthetic datasets when the prior is overestimated. In this case, the total bias approaches the error bound as the sample size grows. In all other settings, including benchmark datasets, the total bias converges to smaller values than the bound, indicating that the theoretical bound is conservative in practice.

## 6 Conclusions

We proposed PU-ECE, the first ECE estimator that can be computed from PU data. Our theoretical analysis provides non-asymptotic bias bounds for PU-ECE and shows that, with an optimal binning strategy, the bias of our estimator converges to the TCE at a rate of $\mathcal{O}((\pi_P^2/n_P + 1/n_U)^{1/3})$. This rate matches the convergence rate of the standard ECE in the supervised setting. We further extended the CMI framework to PU learning, deriving generalization error bounds with the same convergence rate as the bias. Experiments on synthetic and benchmark datasets corroborated the theory: PU-ECE performs comparably to the standard ECE, despite lacking negative labels. These results also reveal two practical caveats: (i) with the number of positive samples fixed, increasing unlabeled data alone does not yield convergence to the true TCE, underscoring the importance of having sufficient positive data for accurate estimation; and (ii) any nonzero prior error makes PU-ECE converge to a biased limit, even though the observed error is often below the theoretical bound.

## Acknowledgements

FF was supported by JST PRESTO Grant Number JPMJPR22C8. MS was supported by JST ASPIRE Grant Number JPMJAP2405.

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
