# OpenReview forum: "Estimating Expected Calibration Error for Positive-Unlabeled Learning"
_TMLR — Accepted by TMLR_

### Review · Reviewer_z4Bc · 2025-08-06

**Summary Of Contributions:**

The paper proposes a novel approach to measure the expected calibration error (ECE) for positive-unlabeled data (PU) to gain more insides into the reliability of a model trained with PU data only.

**Audience:**

Yes

**Audience Explanation:**

The theoretical framework provided is intersting.

**Broader Impact Concerns:**

no broader impact statement required or given

**Claims And Evidence:**

No

**Claims Explanation:**

The authors provide a theoretical framework for estimating ECE in a setting where only positive labeled and unlabeled data is given.

However, the proposed framework is limited to binary classification tasks, where the actual usability of ECE is questionable.
For a two-class problem, the desired confidence is typically 100% for a correct prediction and 0% for an incorrect one. Consequently, the binning strategy inherent to ECE may not be well-suited to this problem and could provide a misleading measure of calibration. This choice of metric requires a more thorough justification.

**Requested Changes:**

Requested Changes:

Please provide formal definitions for the variables n and B immediately following Equation (1).

Open Questions:
1. The authors should provide a justification for the choice of using Softmax as the output activation function, especially considering its potential limitations for the specific task at hand.
2. The applicability of the proposed approach appears to be very limited, as all experiments are only conducted on binary classification tasks. The authors should discuss the challenges of extending their method to a multi-class setting.
3. The use of Expected Calibration Error (ECE) for a binary classification problem is questionable. For a two-class problem, the desired confidence is typically 100% for a correct prediction and 0% for an incorrect one. Consequently, the binning strategy inherent to ECE may not be well-suited to this problem and could provide a misleading measure of calibration. This choice of metric requires a more thorough justification.

---

> ### Author Response · Authors · 2025-08-07
> **Author Response to Reviewer z4Bc**
>
> We thank the reviewer for the constructive and detailed feedback. Below we address each point in turn.
>
> **Q1.  Why restrict the study to binary PU classification?**
>
> A1. Binary is the **de facto standard** setting in both theory and practice. Most existing research focuses on the binary setting, while research on multiclass PU learning remains scarce. Therefore, focusing on the binary PU setting is consistent with common practice and addresses many real-world applications.
>
>
> **Q2.  Can PU-ECE be extended to the multi-class case?**
>
> A2. Yes. Although multiclass PU learning is not the mainstream focus in the literature, our proposed PU-ECE can be naturally extended to some multiclass calibration metrics. For example, it can be easily adapted to classwise-ECE (Kull et al., 2019), which evaluates calibration per class. While other multiclass metrics such as top-1 ECE (Guo et al., 2017), may not be directly compatible, they often reduce the multiclass calibration problem to a one-dimensional evaluation based on the maximum predicted score. In this sense, our research provides an essential first step toward calibration in multiclass PU settings. We will add a short appendix discussing this extension.
>
> **Q3. Is ECE informative for a binary task where the “ideal” confidence is 0 or 1 ?**
>
> A3. Theoretically, the ideal confidence is 0 or 1 only when a classifier achieves 100% perfect accuracy. However, this is rarely the case even in binary classification. For example, accuracy of state-of-the-art models on NLP datasets such as IMDb is ~97% (Liu et al., 2019), and that on physics datasets such as JETCLASS is ~85% (Qu et al., 2022). In such realistic settings, calibration remains highly informative. Furthermore, as our method estimates conditional probabilities, it allows plotting reliability diagrams to visualize where a model is overconfident or underconfident. This contributes to interpretability and model debugging, which we believe adds practical value to using ECE in the PU setting. We will add a short appendix discussing this extension.
>
> **Q4. Justification for the choice of using activation function (softmax/sigmoid)**
>
> A4. Our framework does not require a specific activation function. It only assumes that the confidence score $f(X)$ maps into $[0,1]$ and is absolutely continuous. While sigmoid (binary) and softmax (multiclass) are standard, any monotonic increasing mapping—such as probit or complementary log-log—would equally satisfy the theoretical requirements. We will clarify this generality in Section 2.
>
> **Q5. Minor revision: definitions of $n$ and $B$ (Eq. 1)**
>
> A5. Thank you for catching this. We will replace $n$ with $n_\mathrm{e}$ and add **“$n_\mathrm{e}$ denotes the number of evaluation samples, and $B$ is the number of bins.”**
>
>
> **References:**
> - Kull, Meelis, et al. (2019). _Beyond Temperature Scaling: Obtaining Well-Calibrated Multi-Class Probabilities with Dirichlet Calibration_. NeurIPS.
> - Guo, C., et al. (2017). _On Calibration of Modern Neural Networks_. ICML.
> - Liu, Y., et al. (2019). _RoBERTa: A Robustly Optimized BERT Pretraining Approach_. arXiv:1907.11692.
> - Qu, H., et al. (2022). _Particle Transformer for Jet Tagging_. ICML.

---

> ### Comment · Reviewer_z4Bc · 2025-10-30
> **no rebuttal**
>
> As the authors omit to answer any of my concerns and questions. I'm not convinced that the paper should be published in it's current form.

---

> > ### Author Response · Authors · 2025-10-30
> > **To Reviewer z4Bc**
> >
> > I just realized that the comment I posted before the author response period was not visible to everyone due to my mistake. I apologize for this oversight. The comment should have been visible earlier.

---

### Review · Reviewer_wNcX · 2025-08-28

**Summary Of Contributions:**

The authors seem to have taken the methodology of Futami &Fujisawa 2024 (cited in the paper) and extended it from the standard supervised learning regime into the PU learning regime.

The difference in problem setup means that the empirical distribution of positive examples in each histogram bin is not available, and this affects what theoretical results can be derived. A natural alternative estimator is proposed which is appropriate for the problem setup. Theorem 1 analyzes the statistical estimation error and the binning discretization errors of this estimator. Compared to Futami &Fujisawa 2024 Corollary 1 they provide corrections deriving from the different problem setup. The authors also provide a concentration result bounding the deviation of the bias term from the expected bias. (n.b. the 'bias' term is a random varable that depends on the evalation set so knowing how large deviations are plausible for a single evaluation dataset is valuable). Theorem 2 introduces an error analysis based on the assumed known class marginal distribution $\pi_P$ which is important for this problem setup. Theorem 3 presents analysis of the generalization error, parallell to Futami &Fujisawa 2024 Theorem 5. The proof in appendix points out several of the reasons for discrepancies in the formulas.

The authors also provide a numerical section showing that the theoretical results are observed in practice.

**Audience:**

Yes

**Audience Explanation:**

PU learning is an important scenario in applied machine learning, and finite sample bounds on calibration errors can be used both for inference and for guiding study design (inform sample sizes etc).

One statement in the conclusions "if the number of positive samples is fixed and the number of unlabeled samples increases, PU-ECE does not converge to the true TCE" is a good example of what TMLR readers should be interested in learning from the paper.

**Broader Impact Concerns:**

I do not see any problematic ethical implications of this work.

**Claims And Evidence:**

Yes

**Claims Explanation:**

As far as I can determine, the theoretical claims are supported, and the proofs are clear and possible to follow.
The numerical claims are based on reasonable scenarios and evaluation strategies.

While I find some problems in the clarity of the presentation (see requested changes), I find the article clear enough to understand.

**Requested Changes:**

Would strengthen your work:
- Section 2.1 states "Classifiers are trained by the empirical risk minimization" but many other training methods are used in supervised learning. Regularization often falls outside ERM. Distributional robust optimization is used. Modern ML use in-context learnin. The list goes on. your quote is this misleading.
- First equation in section 2.2.2 has a sum and an indicator function, effectively removing the sum. It corresponds to eq (3) in Futami &Fujisawa 2024, but in that case x is different in the conditional expectation and the indicator. This looks like a typesetting mistake. Please clear it up
- Last paragraph page 3 discuss extensions to TCE/ECE, but this has no bearing on the article. Should be put in background (related work) or preferrably just removed.
- Section 4.1 should define what CMI is in this context. For a reader familiar with CMI as defined in information theory, the explanation that "CMI quantifies the amount of information that a model retains about the training data after observing a test sample." is confusing. I think clarity would be improved by moving the definition of fCMI from appendix B to the main text.
- 5.1 Common Setup / Synthetic data. Defining a distribution in terms of P(Y|X) and P(X|Y) instead of P(Y|X) and P(X) is unnecessarily confusing. Even though your presentation is *correct* it is harder to read than it should be.
- 5.2 Motivational Study should be part of the background. All problems should be motivated there, not in the Experiments section. Paragraph 3, starting with "This collapse is likely due to..." does not belong in a main text section. Such speculation should either be addressed with experiments proving it is the case, or it should be placed in a discussion/conclusion/outlook section. By mixing speculation with the experiment setup, it is harder to separate what is well supported science and what is not. The main point you present in section 5.2 also seems to be "PU learning is considerably more prone to miscalibration than standard supervised learning" which seems besides the point if the article given that I understand the research question as "What are the statistical properties of our propsed estimator?"
- I suggest you remove Figure 2 and 3. The text in 5.3 explains very well that you made sensible choices for estimating the TCE, so the figures don't really add to that. Possibly, keep Figure 2 and remove figure 3, since they are equivalent up to a transformation of the x-axis.

---

> ### Author Response · Authors · 2025-09-30
> **Author Response to Reviewer wNcX**
>
> We thank the reviewer for the thoughtful and constructive feedback.
> Below we respond to each requested change and specify the concrete edits we will make in the revised manuscript.
> ### 1) Wording around ERM (Sec. 2.1)
> **Reviewer’s point.** “Classifiers are trained by ERM” is too strong/misleading.
>
> **Our revision.** We will replace the sentence with:
> > A standard approach for training classifiers is to minimize this quantity, a principle known as empirical risk minimization (ERM).
>
>
> ### 2) Notation in the first equation of Sec. 2.2.2
> **Reviewer’s point.** The indicator and sum make the current form look like a typesetting mistake; it should parallel Eq. (3) in Futami & Fujisawa (2024).
>
> **Our revision.** We will correct the definition of $f_\mathcal{I}(X)$ as follows:
> > $f_\mathcal{I}(x) = \sum_{b=1}^B \mathbb{E}\left[f(X)\mid f(X)\in I_b\right] \mathbb{1}{f(x)\in I_b}$.
>
>
> ### 3) Paragraph on extensions to $L_p$​-TCE/ECE (end of Sec. 2.2)
> **Reviewer’s point.** Not relevant to the paper.
>
> **Our revision.** We will remove the paragraph from the main text.
>
>
> ### 4) Use and definition of “CMI” (Sec. 4.1)
> **Reviewer’s point.** The informal description of CMI is potentially confusing; define clearly in this context and consider moving the formal definition from the appendix.
>
> **Our revision.** We will:
> - Rename the first subsection to **“Functional Conditional Mutual Information (fCMI) for PU learning.”**
> - Replace the informal sentence with a precise statement and **introduce formal definitions of $\mathrm{fCMI}$ and $\mathrm{fCMI}_{\mathrm U}$​ in the main text (moved from Appendix B).
> - Use fCMI or CMI-based analysis framework instead of just referring to “CMI.”
> We expect these edits to eliminate ambiguity for readers familiar with the standard usage of the word CMI.
>
>
> ### 5) Synthetic‑data description (Sec. 5.1)
> **Reviewer’s point.** Defining the data‑generating process via $P(Y∣X)$ and $P(X\mid Y)$ is harder to read than using $P(Y\mid X)$ and $P(X)$.
>
> **Our revision.** We will re‑express the generative description in terms of $p(X)$ and $P(Y\mid X)$.
>
>
> ### 6) “Motivational Study” (current Sec. 5.2): placement and speculative text
> **Reviewer’s point.** The motivation should not appear in the Experiments section; the speculative paragraph (“This collapse is likely due to…”) should be removed or moved to a discussion. The main research question is the statistical properties of our estimator; avoid blending speculation with the experimental setup.
>
> **Our revision.** We will **remove the motivational study**, as it neither motivates nor validates our proposed estimator. The Experiments section will then focus strictly on validating the **theoretical properties** of PU‑ECE (bias/convergence), aligning the narrative with our primary research question.
>
>
> ### 7) Figures 2 and 3 (current Sec. 5.3)
> **Reviewer’s point.** The figures are largely redundant with the accompanying text; consider removing, or keeping only one.
>
> **Our revision.** We will **remove Figure 2 and Figure 3**.
>
> Again, thank you for your careful reading and helpful suggestions.

---

### Review · Reviewer_k7h8 · 2025-09-30

**Summary Of Contributions:**

This paper introduces PU-ECE, the first estimator of expected calibration error (ECE) in positive–unlabeled (PU) learning. Since standard ECE requires full labels (positive + negative), it cannot be applied in PU settings. The authors propose to reframe the ECE computation using Bayes’ rule with empirical frequencies from positive and unlabeled data. They provide non-asymptotic bias bounds, show convergence rates that match the supervised case, and extend information-theoretic generalization error analysis (via conditional mutual information) to PU learning. Experiments on synthetic, MNIST, and CIFAR-10 datasets validate the theoretical claims.

Strengths

Addresses an important problem namelycalibration in PU learning.

Provides rigorous bias analysis and convergence guarantees.

Extends fCMI-based generalization analysis to a new setting.


Weaknesses

Novelty is somewhat incremental; PU-ECE is a natural adaptation of supervised ECE.

Heavy reliance on a known class prior, which is unrealistic in practice; robustness to misestimation is not empirically studied.

Experimental evaluation is narrow (only MNIST/CIFAR binary subsets); lacks validation on real PU application domains.

Limited ablations (no empirical comparison of UMB vs UWB binning).

Practical interpretability is under-discussed (e.g., calibration diagrams, downstream decision costs).

**Audience:**

Yes

**Audience Explanation:**

he work will interest researchers studying calibration, weak supervision, and uncertainty estimation. Extending calibration theory to PU settings fills a genuine gap..

**Claims And Evidence:**

No

**Claims Explanation:**

The theoretical claims are well supported with rigorous derivations and proofs. The convergence rates and bias bounds are clearly demonstrated.
However, the empirical support is thin where results are confined to small-scale datasets and controlled settings. The claim of “comparable performance to supervised ECE” is not convincingly validated beyond synthetic and toy cases. The assumption of a known prior is a significant limitation that weakens the practical strength of the claims.

**Requested Changes:**

Add experiments testing PU-ECE under estimated priors (e.g., +-10% error) to evaluate robustness.

Include at least one real-world PU dataset (e.g., bioinformatics, medical, or text classification) to demonstrate practical relevance.

Provide empirical comparison of UMB vs UWB binning strategies.

Discuss or simulate extensions to multiclass PU calibration.
Add reliability diagrams or calibration plots comparing PU-ECE with supervised ECE.
 Clarify computational complexity and scalability relative to standard ECE.

---

> ### Author Response · Authors · 2025-10-14
> **Author Response to Reviewer k7h8**
>
> We thank the reviewer for the insightful and helpful feedback. Below, we respond to each point and outline the concrete changes that will be incorporated into the revised version of the manuscript.
>
> > Add experiments testing PU-ECE under estimated priors (e.g., +-10% error) to evaluate robustness.
>
> We will add Section 5.3 that experiments the effect of prior estimation error. The results confirmed the theory that the PU-ECE curves shift by approximately the magnitude of the prior error and preserve the overall trend.
>
> > Include at least one real-world PU dataset (e.g., bioinformatics, medical, or text classification) to demonstrate practical relevance.
>
> We will add experiments on the DDI dataset [1], which is a medical text dataset of drug-drug interactions to Sections 5.2 and 5.3.
>
> > Provide empirical comparison of UMB vs UWB binning strategies.
>
> Following your suggestion, we conducted additional experiments comparing UMB and UWB.
> We found that while UWB shows more stable performance on some datasets, UMB performs better on others. Therefore, we will keep UMB as the default binning strategy for consistency with the standard ECE definition, and we add the comparison results to Appendix I.
>
> > Discuss or simulate extensions to multiclass PU calibration.
>
> We have discussed the extension to multiclass PU calibration in Appendix H of the latest revision.
>
> > Add reliability diagrams or calibration plots comparing PU-ECE with supervised ECE.
>
> We have the reliability diagram for PU learning in Appendix G of the latest revision. We will additionally include the reliability diagram of supervised ECE so that we can compare the PU reliability diagram side-by-side.
>
> > Clarify computational complexity and scalability relative to standard ECE.
>
> We will add the sentence "The computational complexity of PU-ECE is $\mathcal{O}(n_\mathrm{U} + n_\mathrm{P})$ given the bins, which is the same as that of the standard ECE." to Section 3.1.
>
> **References**
> [1] Herrero-Zazo, María, Isabel Segura-Bedmar, Paloma Martínez, and Thierry Declerck. 2013. “The DDI Corpus: An Annotated Corpus with Pharmacological Substances and Drug–Drug Interactions.” _Journal of Biomedical Informatics_ 46 (5): 914–920.

---

### Decision · Action_Editor_JFqy · 2025-12-02

**Recommendation:** Accept with minor revision

**Additional Comments:**

There were several small suggestions regarding typos and making the discussion clearer. For instance:

**Reviewer k7h8**
- “Add experiments testing PU-ECE under estimated priors (e.g., ±10% error) to evaluate robustness.”
- “Include at least one real-world PU dataset… to demonstrate practical relevance.”
- “Provide empirical comparison of UMB vs UWB binning strategies.”
- “Discuss or simulate extensions to multiclass PU calibration.”
- “Add reliability diagrams or calibration plots comparing PU-ECE with supervised ECE.”
- “Clarify computational complexity and scalability relative to standard ECE.”


**Reviewer wNcX**
- “Section 4.1 should define what CMI is… move fCMI from appendix B to the main text.”
- “Remove speculative paragraph (‘This collapse is likely due to…’).”
- “5.2 Motivational Study should be part of the background… not experiments.”
- “Remove Figure 3… possibly keep Figure 2.”
- “Last paragraph page 3… should be moved or removed.”
- “First equation in section 2.2.2… looks like a typesetting mistake.”
- “‘Classifiers are trained by ERM’… is misleading.”
- “Defining P(Y|X) and P(X|Y)… is unnecessarily confusing.”

**Reviewer z4Bc**
- “The authors should discuss the challenges of extending their method to a multi-class setting.”
- “Provide a justification for the choice of using Softmax… especially considering its potential limitations.”
- “This choice of metric requires a more thorough justification.”
- “Provide formal definitions for the variables n and B immediately following Equation (1).”

**Audience:**

Yes

**Audience Explanation:**

The topic studied in this work --- how to accurately quantify uncertainty in classification tasks under the presence of positive unlabelled data --- is an important research direction of interest to the TMLR and ML community more broadly.

**Claims And Evidence:**

Yes

**Claims Explanation:**

This work considers the question of calibration in supervised binary classification tasks where unlabelled data from only one of the two classes is available. The work is mostly theoretical in nature, and the main result is a convergence bound between the empirical and population expected calibration errors for a proposed estimator for the calibration that leverages the unlabelled data.

The overall assessment of all the reviewers were positive, with the main strength highlighted being the lack of theoretical results on uncertainty quantification under unlabelled data.

The main weaknesses raised was the incremental nature of the theoretical analysis, which relies on previous technical results for the standard expected calibration error. Other points raised was the limitation to binary settings only and the assumption of the knowledge of a class prior.

The reviewers made many relevant suggestions that could help improving the manuscript, from correcting typos to clarifying the discussion / experiments. I am attaching a summary below. Conditioned on the implementation of these changes I am happy to recommend the paper for publication at TMLR.